# NbMLP43 Ubiquitination and Proteasomal Degradation via the Light Responsive Factor NbBBX24 to Promote Viral Infection

**DOI:** 10.3390/cells12040590

**Published:** 2023-02-11

**Authors:** Liyun Song, Yubing Jiao, Hongping Song, Yuzun Shao, Daoshun Zhang, Chengying Ding, Dong An, Ming Ge, Ying Li, Lili Shen, Fenglong Wang, Jinguang Yang

**Affiliations:** 1Key Laboratory of Tobacco Pest Monitoring, Controlling & Integrated Management, Tobacco Research Institute of Chinese Academy of Agricultural Sciences, Qingdao 266101, China; 2Hubei Engineering Research Center for Pest Forewarning and Management, Agricultural College, Yangtze University, Jingzhou 434025, China

**Keywords:** potato virus Y, resistance, MLP-like protein 43, ubiquitination, ubiquitin–proteasome system, B-box zinc finger protein 24

## Abstract

The ubiquitin–proteasome system (UPS) plays an important role in virus–host interactions. However, the mechanism by which the UPS is involved in innate immunity remains unclear. In this study, we identified a novel major latex protein-like protein 43 (NbMLP43) that conferred resistance to *Nicotiana benthamiana* against potato virus Y (PVY) infection. PVY infection strongly induced NbMLP43 transcription but decreased NbMLP43 at the protein level. We verified that B-box zinc finger protein 24 (NbBBX24) interacted directly with NbMLP43 and that NbBBX24, a light responsive factor, acted as an essential intermediate component targeting NbMLP43 for its ubiquitination and degradation via the UPS. PVY, tobacco mosaic virus, (TMV) and cucumber mosaic virus (CMV) infections could promote NbMLP43 ubiquitination and proteasomal degradation to enhance viral infection. Ubiquitination occurred at lysine 38 (K38) within NbMLP43, and non-ubiquitinated NbMLP43(K38R) conferred stronger resistance to RNA viruses. Overall, our results indicate that the novel NbMLP43 protein is a target of the UPS in the competition between defense and viral anti-defense and enriches existing theoretical studies on the use of UPS by viruses to promote infection.

## 1. Introduction

Plant viral diseases are major constraints to global agricultural production [1,2]. To address this problem, in-depth analysis of pathogenic and host resistance mechanisms can provide significant breakthroughs. In particular, a comprehensive knowledge on the mechanisms of host innate immunity and viral anti-immunity is the basis for understanding and controlling future crop viral diseases.

The ubiquitin–proteasome system (UPS) plays an important role in the arms race between the host’s immunity and the virus’s anti-immunity [3,4,5]. Recently, further studies have revealed that organisms use the UPS to resist pathogens, and pathogens have evolved mechanisms to recruit the UPS against immunity [6,7,8]. Prior to degradation by proteasomes, most UPS substrate proteins are covalently bound to ubiquitin via three energy-dependent steps as follows: (1) ubiquitin-activating enzyme (E1) activation, (2) ubiquitin-conjugating enzyme (E2) conjugation, and (3) ubiquitin ligase (E3) connection [9]. E3 is responsible for the final labeling of proteins and provides specificity for the UPS process [10].

Major latex proteins (MLPs) were first identified in poppy latex [11,12]. MLP-like proteins are encoded by the homologous *MLP* gene, and MLPs can resist pathogenic infection. Transcription in *Arabidopsis thaliana* of *MLP28* (AT1G70830) and *MLP3* were induced by *Alternaria brassicicola* and *Plasmodiophora brassicae* infections, respectively [13,14]. In cotton, *MLP28* responds to *Verticillium dahliae* infection [15,16]. NbMLP28, which we have previously identified, potato virus Y (PVY) infection induces transcriptional upregulation of *NbMLP28*. Silencing of *NbMLP28* renders *Nicotiana benthamiana* more vulnerable to PVY infection, and overexpression of *NbMLP28* inhibits the infection of PVY to *N. benthamiana* [17]. Existing evidence shows that MLP43, the mRNA expression which is upregulated following PVY infection, may also be involved in the antiviral process. However, the involvement of MLP in the mechanism of UPS degradation has not been investigated. Infection by PVY causes significant harm to commercial crops [18,19], and the discovery and functional elucidation of plant resistant genes provides a theoretical basis for viral resistance breeding. The hypersensitive resistance response genes *Ny*, *Nc*, and/or *Nz* are present in many potato varieties and recognize different PVY strains [20]. The TNL immune receptor Ry_sto_ recognizes the PVY coat protein, conferring resistance to PVY [21]. *Ry_chc_* also displays strong resistance to PVY [22]. In this study, the model plant *N. benthamiana* was chosen as the primary research host system. Based on omics data, combing with protein function verification technology, verification experiments were carried out through the construction of transgenic plants to clarify the function of the identified MLP43 (denoted as NbMLP43) and reveal the mechanism of its degradation. This study investigated the molecular mechanism of NbMLP43 degradation for the first time, extending the current knowledge of how viruses exploit UPS for anti-host immunity.

## 2. Materials and Methods

### 2.1. Plant Materials and Viral Strains

Four sets of *N. benthamiana* plants were used: (a) wild-type (seeds preserved in our laboratory); (b) *NbMLP43*-overexpressing (OE) transgenic plants with a red florescent protein (RFP) tag constructed for this study by Wuhan Leaf Power Biotechnology Company Limited (Wuhan, China); (c) *mlp43*, a CRISPR/Cas9-based knockout of *NbMLP43*-transgenic *N. benthamiana* constructed by Wuhan Towin Biotechnology Company Limited (Wuhan, China); and (d) NbMLP43(K38R), constructed with a mutation of the ubiquitination site K38 to R38. All these plants were grown in a chamber with a humidity of 50–60% and a 16-h/8-h light/dark photoperiod at 25 °C. The following viruses were used in these analyses: CMVI B, TMV U1, PVYN:O strains, and PVY expressing the green fluorescent protein (PVY-GFP) [23].

### 2.2. Vector Construction

Two expression vectors were used, each harboring a fusion gene encoding the red fluorescent protein (RFP) tag: p35S::MLP43-RFP and p35S::MLP43(K38R)-RFP. The construction of these two vectors involved the homologous recombination of the coding sequence (CDS) of NbMLP43 or NbMLP43 (K38R; point mutation of lysine 38 to arginine) with the intermediate vector pFu46-RFP, and the pEarleygate100 vector was ligated with the vector via the LR reaction as described previously [23]. The p35S::BBX24-GFP expression vector was constructed through homologous recombination of the CDS of NbBBX24 with the intermediate vector pFu28-GFP, followed by ligation to the pEarleygate100 vector via the LR reaction [23].

The pNC-Enc-MLP43 vector was constructed through homologous recombination of the CDS of NbMLP43 with the vector pNC-Enc; the pNC-Enn-BBX24 vector was constructed through homologous recombination of the CDS of NbBBX24 with the vector pNC-Enn.

Four gene silencing vectors (pTRV2::NPR1, pTRV2::COI1, pTRV2::EIN2, and pTRV2::BBX24) were constructed by selecting 200–300 bp fragments from the corresponding CDS and inserting them into the pTRV2 vector through in-fusion technology [23]. Additional vectors (pBD-MLP43, pBD-BBX24, pAD-MLP43, pAD-CP, pAD-HC-Pro, pAD-NIa, pAD-NIb, pAD-VPg, pAD-P1, pAD-P3, pAD-6K1, pAD-6K2, and pAD-CI) were constructed through recombination of the corresponding CDS to the vector pADT7 or pBKT7.

### 2.3. RNA-Sequencing and Transcriptional Analysis

Total RNA from PVY-inoculated and PBS-treated *N. benthamiana* was extracted with the TRIzol reagent (Vazyme, Nanjing, China). The obtained mRNA was enriched by using magnetic beads with Oligo(dT) to combine with m-ploy A tail and cut into short fragments. Single-stranded cDNA was synthesized using mRNA as a template, followed by second-strand synthesis, and then double-stranded cDNA was purified using AMPure XP beads. The purified double-stranded cDNA was subjected to end repair, added with A tail, and connected to the sequencing adapter. Then we used AMPure XP beads for fragment size selection and PCR enrichment to obtain the final cDNA library. The resulting PCR products were purified, and the quality of the libraries was assessed using the Agilent Bioanalyzer 2100 system. After the insert size was as expected, the effective concentration of the library was accurately quantified (the effective concentration of the library >2 nM) to ensure the quality of the library using the Q-PCR method. Different libraries were then pooled according to the requirements of effective concentration and target offline data volume, and HiSeq sequencing was performed.

The clean reads were aligned individually against the reference genome of *N. benthamiana* (https://solgenomics.net/organism/Nicotiana_benthamiana/genome, 28 May 2018). The number of clean reads mapped to a specific gene were counted and then presented as fragments per kilobase of transcript per million mapped reads (FPKM) using Cufflinks [24]. Differentially expressed genes (DEGs) between the PVY-infected and the PBS-treated control samples (Mock) were identified using the DESeq software [25]. The DEGs were ranked individually based on the average log_2_ Fold Change (FC) and the false discovery rate (FDR) Q values and then filtered using FDR Q < 0.05 and |log_2_ FC| ≥1. Gene function was annotated according to the following databases: Gene Ontology (GO), Kyoto Encyclopedia of Genes and Genomes (KEGG) Ortholog (KO), Swiss-Prot, and NCBI non-redundant protein sequences (Nr). GO enrichment of DEGs was determined based on the Wallenius noncentral hypergeometric distribution using the GOseq R package [24].

### 2.4. Cloning and Sequence Analysis of NbMLP43

Fresh *N. benthamiana* leaf tissue (100 mg) was prepared, and TRIzol reagent (Vazyme, Nanjing, China) was used to extract RNA. Total RNA (2 μg) and reverse transcriptase (Vazyme) were used to synthesize cDNA. According to the *N. benthamiana* genome data at the Sol Genomics Network, the open-reading frame sequence of MLP43 was amplified for the first time from *N. benthamiana* (denoted as NbMLP43) using primers MLP43F/MLP43R, and the sequence was submitted to the National Center for Biotechnology Information under the accession number MK780770. SMART was used for domain prediction. SWISS-MODEL was used for protein 3D structure prediction [26]. MEGA7 was used to generate a phylogenetic tree [27]. MLP43 proF and MLP43 proR primers were used to amplify the 2000-bp promoter sequence, and the analysis of potential cis-regulatory elements in the *NbMLP43* promoter was performed using the online program Plant CARE (http://bioinformatics.psb.ugent.be/webtools/plantcare/html, 27 May 2018). The primers used for plasmid construction are listed in Appendix A.

### 2.5. Virus-Induced Gene Silencing (VIGS)

For TRV-VIGS assays based on the tobacco rattle virus (TRV) VIGS system, the constructed silencing vector was transformed into *Agrobacterium tumefaciens* LBA4 (Biomed, Beijing, China) and mixed with pTRV1 at a 1:1 ratio to infiltrate 3-week-old *N. benthamiana* (OD600 = 0.8). pTRV1: pTRV2 was the negative control, whereas pTRV1:pTRV2-PDS was the positive control [28]. The gene silencing efficiency was determined after 14 days. To determine the effects of VIGS on the viral infection, PVY was inoculated on the 15th day, and samples were collected at selected time points to detect the relative expression of PVY *CP*.

### 2.6. GFP and RFP Fluorescence Detection

A Leica SP8 confocal microscope (Leica Microsystems, Shanghai, China) was used to analyze the subcellular distribution of NbMLP43 in NbMLP43-OE plants with an RFP tag, with or without PVY-GFP. To visualize the subcellular distribution of NbMLP43 in single cells, we used cellulase R-10 (Yakult, Tokyo, Japan) and dissociative enzyme R-10 (Yakult, Japan) to prepare the enzymatic solution, lysed the leaf tissue in the dark for 3 h, then centrifuged to remove the supernatant, and resuspended to prepare protoplasts [26]. For the subcellular distribution test, a 25 mW, 488 nm laser was used to excite GFP, and signals were captured within the 495–535 nm wavelength range. RFP was excited using a 25 mW, 552 nm laser, and signals were captured within the 580–630 nm wavelength range. The 488 and 552 nm lasers were used to scan successive images (20 μm × 20 μm) at a scanning interval of 1.0 s [23]. To verify the effect of *NbMLP43* knockout and OE in plants and visually detect the accumulation of virus in the inoculated leaves, we infiltrated tobacco leaves with PVY-GFP and compared its fluorescence with that of the control plants under a handheld ultraviolet lamp. Only one leaf per plant was inoculated, and these experiments were repeated three times, with each experiment containing three biological replicates.

### 2.7. Nitroblue Tetrazolium (NBT) and Diaminobenzidine (DAB) Staining

H_2_O_2_ and O2^−^ were detected with DAB and NBT, respectively [29,30]. DAB and NBT Staining solutions were prepared according to the manufacturer’s instructions (Solarbio, Beijing, China), and fresh leaf tissue was added after gentle shaking for 5 h in the dark. During this time, a decolorizing solution containing acetic acid, ethanol, and glycerol (1:3:1) was used. The samples were then decolorized in a boiling water bath for 3 min. The leaves were stored in a protective solution of ethanol:glycerin (4:1) and photographed for observation.

### 2.8. Hormone Spraying Test and Determination of Salicylic Acid (SA) Content

Four-week-old wild-type *N. benthamiana* was grown in an artificial climate chamber, and the leaves were sprayed with either 0.5 mM SA, 0.1 mM methyl jasmonate (Me-JA), or 0.05 mM ethephon (containing 0.02% Tween 20). The control treatment consisted of *N. benthamiana* sprayed with water containing 0.02% Tween 20. Samples were collected 24 h after treatment, quickly frozen in liquid nitrogen, and stored at −80 °C until further analysis.

The *mlp43* and NbMLP43-OE plants were infiltrated with PVY-GFP, and the SA content of the treated leaves was determined after 4 days. Control leaves were infiltrated with phosphate-buffered saline. Individuals ground about 50 mg of leaf tissue in liquid nitrogen, added 400 μL of methanol extract containing 1% acetic acid to extract, and then performed quantitative detection of SA with liquid chromatography and mass spectrometry [31]. This work was conducted by Wuhan Punais Biology Technology Co., Ltd. (Wuhan, China). Only one leaf per plant was inoculated, and these experiments were repeated three times, with each experiment containing three biological replicates.

### 2.9. MG132 and 3MA Treatment

Consistent growth of *N. benthamiana* was ensured using four-week-old plants, and MG132 (or 3MA) and the control reagent dimethyl sulfoxide were diluted to the same proportions. The lower surface (the abaxial surface) of the leaves were infiltrated with a needleless syringe, and the same position of the leaves was selected for each treatment [32]. These experiments were repeated three times, with each experiment containing three biological replicates. After 2 h, the leaves were inoculated with PVY (diluted 60 times with PBS), and samples were collected 24 and 48 h after inoculation and stored in a refrigerator at −80 °C for later use.

### 2.10. RT-qPCR

Total RNA extraction and cDNA synthesis were performed as described above. qPCR was performed using the SYBR Premix Ex TaqTM kit according to the manufacturer’s instructions (Vazyme) on an ABI 7500 fast real-time quantitative PCR instrument. The expression level of the *N. benthamiana actin* gene (*NbActin*, AY179605) was used as internal control. The PVY-F/PVY-R and β -ActinQF/β -ActinQR primers (Appendix A) were used to amplify PVY *CP* and *β-actin*, respectively. Real-time quantitative primers TMV-F/TMV-R and CMV-F/CMV-R were used to detect the relative expression of TMV *CP* and CMV *CP* in different treatments, respectively. Relative expression levels of these genes were calculated using the 2^−ΔΔCT^ method. These experiments were repeated three times, with each experiment containing three biological replicates for all the RT-qPCR experiments.

### 2.11. Yeast Two-Hybrid (Y2H) Assay

The constructed pBD-MLP43 plasmid was transformed into the Y2H Gold strain to obtain the bait strain, and the Y2H screening test was performed using the Y187 yeast library. The Y187 yeast library and the bait strain were mixed for mating experiments and incubated at 30 °C for 24 h in a shaker incubator at 40 rpm. Mating cells were collected and transferred to SD/-Leu/-Trp (DDO) plates and DDO/x-a-gal/Aba (DDO/X/A) screening plates for preliminary low-pressure screening. Suspected blue colonies growing on DDO/X/A plates were transferred to QDO/X/A plates and incubated at 30 °C for more than 5 days for further high-pressure screening. Positive clones were selected for PCR amplification and sequencing for identification.

For the single-point interaction verification assay, 200 ng each of pBD-Bait plasmid and pAD-prey plasmid were co-transformed in Y2H Gold-competent cells and plated on DDO/X/A (200 ng/mL) plates. When the yeast grew after 3–5 days, a single colony was selected and transferred onto a QDO/X/A (200 ng/mL) plate. Interaction between the foreign protein and target was verified based on the growth of each transformation combination on this plate.

### 2.12. Co-Immunoprecipitation (Co-IP) and Western Blotting

The constructed p35S::MLP43-RFP and p35S::BBX24::GFP expression plasmids were subsequently transferred into *Agrobacterium tumefaciens* LBA4404. Briefly, *Agrobacterium tumefaciens* was suspended in 10 mmol/L MES, 200 μmol/L AS, and 10 mmol/L MgCl2 and mixed at a ratio of 1:1, when an OD600 of 0.8 was achieved. A needle-free syringe was used to infiltrate the lower surface (the abaxial surface) of 4-week-old *N. benthamiana* leaves, which had been inoculated with PVY for 4 h. The p35S::00::RFP and p35S::BBX24::GFP expression vectors were used as controls. Samples were collected 3–4 days after treatment to extract the total plant protein that was purified using magnetic beads coupled with an anti-GFP antibody (ABclonal, Wuhan, China). SDS-PAGE and immunoblotting with an anti-RFP antibody (ab62341, Abcam, Cambridge, UK) were performed to verify the protein interaction between NbMLP43 and NbBBX24 based on the size difference between the treatment and control protein bands.

To determine PVY, TMV, and CMV protein content, anti-PVY CP-specific (SRA20001, Agdia, Elkhart, IN, USA), anti-TMV CP (SRA57400, Agdia, Elkhart, IN, USA), anti-CMV CP (SRA44501, Agdia, Elkhart, IN, USA), and anti-β-actin antibodies (ABclonal, Wuhan, China) were used for western blotting. The ubiquitination level of NbMLP43 was detected using an anti-ubiquitin antibody (PTM-1107, PTM BIO, Hangzhou, China).

### 2.13. Bimolecular Fluorescence Complementation (BIFC) Assay

The constructed pNC-Enc-MLP and pNC-Enn-BBX24 plasmids were subsequently transferred into *Agrobacterium tumefaciens* GV3101. We used the above suspension to suspend pNC-Enc-MLP43 and pNC-Enn-BBX24 *Agrobacterium tumefaciens* separately, then mixed and infiltrateed *N. benthamiana* at a ratio of 1:1, and observed the presence or absence of YFP signal with a laser confocal microscope after 48 h.

### 2.14. Statistical Analysis

Data are expressed as mean ± standard deviation of at least three independent experiments. Duncan’ s multiple range test analysis of variance (ANOVA) and independent sample *t*-test were performed using SPSS (v21, IBM, Armonk, NY, USA). Statistical significance was set at *p* < 0.05.

## 3. Results

### 3.1. Identification and Sequence Analysis of NbMLP43

PVY is a member of the genus *Potyvirus* and severely threatens crop yield and quality. To explore plant defense mechanisms against PVY, RNA-sequencing based transcriptome analysis was performed. Transcriptomic analysis of PVY-infected tobacco plants yielded 61,495 transcripts, 1060 of which were upregulated and 1935 downregulated (Appendix A). Based on Gene Ontology term analysis, the upregulated genes were involved in the following biological processes: regulation of primary metabolic processes, stress responses, abiotic stimulus responses, and defense responses to viral infection (Appendix A). Upregulated novel gene encoding NbMLP43 was identified to be involved in defense responses.

A phylogenetic tree was constructed to compare NbMLP43 with 13 isolates of MLP family members of related species, NbMLP43 and NbMLP28 isolates clustered into one closely related group, supporting the hypothesis that the NbMLP43 was involved in PVY infection (Appendix A) [17]. Multiple alignment analysis of the protein sequences further revealed low similarities between *N. benthamiana* NbMLP28 and *Arabidopsis* MLP43, although both contained a Gly-rich loop with the GXXXXXG sequence (Appendix A). The NbMLP43 protein structure was similar to the 3D structures of NbMLP28 and *Arabidopsis* MLP43 (Appendix A–D). SMART also predicted that NbMLP43 contains the Bet v 1 domain (2-146 aa), suggesting its function in defense response. The 2000-bp long *NbMLP43* promoter sequence was cloned and analyzed, and we found that it contained cis-acting elements related to SA, light, stress, low temperature, and auxin (Table 1).

### 3.2. NbMLP43 Regulates SA Signaling via a Feedback Regulatory Mechanism

RT-qPCR results showed that *NbMLP43* was expressed in various tissues of wild-type *N. benthamiana*, and the level of *NbMLP43* transcripts in leaf tissues was higher than that in the other tissues tested (Figure 1A). Viral inoculation experiments showed an upward trend in viral accumulation at 1, 3, 5, and 7 days after PVY inoculation (dpi) (Figure 1B). Similarly, PVY infection induced the upregulated expression of *NbMLP43* at 1 dpi and reached a maximum at 3 dpi (Figure 1C). Laser confocal results showed that NbMLP43 was mainly distributed in the cytoplasm and nucleus, with or without viral infection (Figure 1G); these results were further verified using prepared protoplasts (Appendix A).

To probe the hormonal pathway of *NbMLP43*, its transcriptional levels were determined in *N. benthamiana* plants treated with either 0.5 mM SA, 0.1 mM Me-JA, or 0.05 mM ethephon. All three treatments increased the transcription level of *NbMLP43*, with SA treatment inducing the largest increase of 1.9 times that of the control (Figure 1D). VIGS was used to silence the key signaling genes *NPR1*, *COI1*, and *EIN2* of SA, JA, and Ethylene (ET), respectively, in wild-type *N. benthamiana* to further analyze the relationship between *NbMLP43* expression and the SA, JA, and ET signaling pathways. The silencing efficiencies of these genes were 72%, 70%, and 75%, respectively (Appendix A). *NbMLP43* expression was downregulated by 32% following NPR1 silencing; however, this effect was not so pronounced following *COI1* and *EIN2* silencing (22% and 23%, respectively; Figure 1E). The SA content was increased following NbMLP43 overexpression (Figure 1F).

### 3.3. Knockout and Overexpression Experiments Reveal the Antiviral Function of NbMLP43

No phenotypic differences were observed between the *mlp43* mutant and control plants (Figure 2C). PVY CP expression in *mlp43* was considerably higher than that in the control group at 1, 3, 5, and 7 dpi; particularly, it was 5.6 and 11.5 times higher than that in the control group at 5 and 7 dpi, respectively (Figure 2A). Western blot analysis revealed that the level of PVY CP in *mlp43* was higher than that in the wild-type at 5 dpi (Figure 2B). At 12 dpi, a green fluorescent protein (PVY-GFP) signal was detected in both *mlp43* and wild-type leaves, but the infected area in *mlp43* system leaves was higher than that in wild-type leaves (Figure 2c). Reactive oxygen species (ROS) are essential signaling molecules in plant responses to stress, especially viral infection. ROS levels in PVY-inoculated plants were assessed at 5 dpi using DAB and NBT staining. Inoculation with PVY-GFP resulted in the accumulation of H_2_O_2_ (stained with DAB) and O2^−^ (stained with NBT) in *mlp43* and WT plants (Figure 2D). However, greater H_2_O_2_ accumulation and O2^−^ production were observed in *mlp43* plants compared with WT plants (Figure 2D). Taken together, *mlp43* plants showed high sensitivity to PVY infection.

NbMLP43-OE seedlings were also inoculated with PVY-GFP, and the difference in the expression of PVY between the NbMLP43-OE and wild-type plants was detected based on the mRNA and protein levels of PVY CP and GFP fluorescence intensity. The RNA expression of PVY CP showed that the NbMLP43-OE plants had a lower viral content than the wild-type plants at 1, 2, 3, and 4 dpi (Figure 2E). The sample from 4 dpi was subjected to western blotting, the results of which were consistent with the gene-level results (Figure 2F). The difference in PVY-GFP fluorescence between the NbMLP43-OE and wild-type plants was also observed at 12 dpi, and the GFP signal in systematic leaves of NbMLP43-OE plant was significantly lower than that of wild type (Figure 2G). These results indicated that *NbMLP43* overexpression inhibited PVY infection and that *NbMLP43* negatively regulated the expression of the virus, exhibiting antiviral functions.

### 3.4. NbMLP43 Degradation via the UPS Pathway Promotes Viral Infection

PVY infection highly induced *NbMLP43* transcription (Figure 1C) but decreased its protein level (Figure 3A). To verify the hypothesis that NbMLP43 is degraded via the UPS or autophagy, proteasome, or autophagy pathway inhibition treatments were performed using MG132 and 3MA, respectively. NbMLP43 levels were higher in MG132-treated plants than in control (DMSO) plants, whereas 3MA treatment did not affect the amount of NbMLP43. These results imply that NbMLP43 was degraded from the UPS pathway rather than from autophagy (Figure 3B).

To clarify the role of the UPS in host–virus interactions, we examined the trend in viral expression at 1, 3, 5, and 7 dpi at the protein level. Western blotting with a pan anti-ubiquitin antibody was performed to determine the ubiquitination levels in the infected leaves (Figure 3D). The results showed that ubiquitination levels were highest at 5 dpi (Figure 3D). Treatment with the proteasome inhibitor MG132 significantly downregulated PVY expression at both the transcriptional and translational levels compared with the control. The transcriptional level of PVY CP was 59% and 57% lower than that of the control after 24 and 48 h of MG132 treatment, respectively (Figure 3E). Western blotting results showed that the virus content was much lower in MG132-treated plants, even after 48 h (Figure 3F). These results showed that the host UPS was inhibited and the expression of PVY CP was significantly reduced, further verifying that the UPS plays an important role in the interaction between the virus and the host.

To probe the effect of NbMLP43 ubiquitination on viral infection, we conducted a quantitative study of ubiquitination proteomics using ubiquitination enrichment technology, coupled with liquid chromatography and tandem mass spectrometry. NbMLP43 ubiquitination increased at the K38 site (Appendix A). Lysine (K38) was mutated to arginine (R38), and a pull-down assay was used to purify NbMLP43, carrying an RFP tag for subsequent verification experiments. The ubiquitination level of NbMLP43 increased following PVY infection, but it decreased following mutation of the ubiquitinatiosite (Figure 3G). Validation experiments were performed using NbMLP43(K38R) plants with ubiquitination site mutation and NbMLP43-OE. The ubiquitination level of NbMLP43 was down-regulated in NbMLP43(K38R) plants (Figure 3G), as well as the expression of PVY CP at the gene and protein levels (Figure 3H,I). Host ubiquitination participated in PVY infection, and NbMLP43 ubiquitination was beneficial to PVY infection. Thus, we could infer that NbMLP43 was degraded via the UPS pathway to promote viral infection.

### 3.5. Screening and Identification of the Interaction between NbBBX24 and NbMLP43

To investigate the degradation mechanism of NbMLP43 through the UPS pathway and identify the E3 ubiquitin ligase specifically targeting NbMLP43, a Y2H assay was carried out with NbMLP43 as the bait protein, which identified NbBBX24 (Appendix A). To verify the interaction between NbMLP43 and NbBBX24, we performed Y2H point-to-point verification and Co-IP assays. The results showed that the positive control grew, the negative control did not grow, and the NbMLP43- and NbBBX24-co-transformed strains grew on SD/-Ade/-His/-Leu/-Trp/X/A (QDO/X/A) plates; neither BD-NbBBX24 nor AD-NbMLP43 underwent self-activation, indicating an interaction between NbMLP43 and NbBBX24 (Figure 4A and Appendix A). p35S::MLP43-RFP and p35S::BBX24-GFP co-infiltrated *N. benthamiana*. The subcellular localization of these two proteins was observed after 48 h; these two proteins were expressed, and all were distributed in the nucleus and cytoplasm (Figure 4C). Co-IP analysis further verified the interaction between these two proteins (Figure 4B). Protein samples co-expressed with NbMLP43 and NbBBX24 were blotted with anti-GFP or anti-RFP antibodies, serving RFP as a negative control. The results showed that NbMLP43, NbBBX24, and RFP were all expressed (Figure 4B Input). The total protein was purified using magnetic beads carrying an anti-RFP antibody, and the eluted protein samples were RFP-containing proteins and plant endogenous proteins interacting with NbMLP43, the results showed that NbBBX24 was only precipitated in the presence of NbMLP43 (Figure 4B IP). In addition, we used BIFC to verify the in vivo interaction between NbMLP43 and NbBBX24. YFP signal was detected in leaves co-infiltrated with pNC-Enc-MLP43 and pNC-Enn-BBX24 (Figure 4D). These results further confirmed the interaction between NbBBX24 and NbMLP43.

### 3.6. NbBBX24 Regulates Viral Infection Photoperiodically and via the UPS Pathway

Follow-up studies were conducted to explore whether NbBBX24 was involved in NbMLP43 ubiquitination and viral infection. RT-qPCR was performed to confirm whether the light signal can induce *NbBBX24* expression, and the treatments with light/dark photoperiods of 16/8 h were 1.48 and 4.13 times those of the control (light/dark photoperiods of 8/16 h) at 2 and 4 dpi, respectively (Figure 5A). The viral content of the two photoperiod treatment groups was detected under the influence of inoculating PVY, and the results showed that light promoted viral infection at transcript level by 13 times and 7.85 times at 2 and 4 dpi, respectively (Figure 5B). The protein levels were consistent with these findings at 4 dpi (Figure 5D). RFP-tagged MLP43-OE plants were treated with light/dark photoperiods of 16/8 h and 8/16 h. At 4 dpi, light treatment increased the ubiquitination of NbMLP43, while PVY CP expression was upregulated (Figure 5D). After silencing *BBX24*, the above two photoperiods were treated at 2 and 4 dpi; the results showed that the viral content was not significantly up-regulated at the transcriptional and protein levels (Figure 5C,E).

*NbBBX24* expression following PVY infection was also determined at the gene level; the expression was 4.8 times higher following PVY infection than that of the control group (Figure 5F). PVY CP expression, determined at the gene and protein levels, was downregulated following *NbBBX24* silencing (Figure 5H–I). NbMLP43 ubiquitination was decreased, whereas NbMLP43 expression was increased (Figure 5G).

To verify the hypothesis that *NbBBX24* expression is specifically induced by PVY,ž, the interaction between the 10 proteins encoded by the PVY genome (namely, CP, HC-Pro, NIa, NIb, 6K1, 6K2, VPg, CI, P1, and P3) and NbBBX24 was assessed using a yeast point-to-point verification assay. There was no direct interaction between PVY-encoded proteins and NbBBX24 (Figure 6A). Both TMV and CMV infections promoted NbMLP43 ubiquitination and downregulated NbMLP43 expression (Figure 6B). These infections also induced NbBBX24 expression (Figure 6C). Furthermore, TMV- or CMV- infected NbMLP43 (K38R) and NbMLP43-OE plants, respectively, with NbMLP43 (K38R) plants significantly inhibited viral infection at both the gene and protein levels (Figure 6D–G). These results indicated that RNA viral infections could nonspecifically induce the ubiquitination-mediated degradation of NbMLP43 in host seedlings to enhance self-infection and that non-ubiquitinated NbMLP43 (K38R) conferred stronger resistance to RNA viruses.

## 4. Discussion

The UPS plays an important role in virus–host interactions [33,34,35]. Several studies have revealed the mechanism with which the UPS mediates the degradation of viral proteins to inhibit viral infection, such as through replicase, CP, MP, and γ b proteins [6,7,36,37]. Viruses recruit the UPS to degrade functional proteins and promote self-infection [38,39,40]. In this study, we identified a novel protein, NbMLP43, that conferred resistance to PVY infection in *N. benthamiana*. PVY infection induced *NbMLP43* expression at the transcriptional level; however, viral infection induced its ubiquitination and degradation. Furthermore, we identified that the light response factor NbBBX24 interacted with NbMLP43 and was involved in its ubiquitination and degradation via the UPS pathway. Overall, we have clarified the molecular mechanism underlying NbMLP43 degradation. Our findings serve as a guide to future studies investigating the response of UPS to host immunity.

The major latex protein/ripening-related protein (MLP/RRP) subfamily is a class of proteins that plays multiple roles in stress response and belongs to the second-largest family of the Bet v 1 superfamily [41,42]. MLPs may play an important role under biotic and abiotic stresses. Overexpression of *Arabidopsis thaliana* MLP43 remarkably enhanced drought resistance based on the abscisic acid signaling pathway [43]. Cotton MLP28 is a positive response factor for *Verticillium dahliae* resistance based on the ethylene signaling pathway [16]. MLP-PG1 from zucchini (*Cucurbita pepo*) is involved in indirect resistance to plant diseases, especially those caused by fungal pathogens, through inducing *PR* genes [44]. However, the biological function of this protein in the defense against viral infection remains unclear. *MLP* has been also detected in the stem phloem of CMV-infected melon plants [45], and NbMLP28 could inhibit the infection of PVY in *N. benthamiana* and is highly expressed in the jasmonic acid signaling pathway [17]. Given the consistent observation of *MLP*-induced expression in response to viral infection, we examined the biological role of this protein in virus–host interactions. To our knowledge, this is the first study to identify NbMLP43 in the model plant *N. benthamiana*. Functional validation analysis revealed that this novel protein acts as a negative regulator of viral infection; NbMLP43 regulates SA signaling via a feedback regulatory mechanism. Importantly, our study provides a valuable basis for the characterization and expression analysis of MLP genes and new insights into the response of MLPs to biological stress.

The ubiquitin–proteasome and autophagy pathways are the two main protein degradation pathways. The VPg protein of turnip mosaic virus mediates the SGS3 degradation through the 20S ubiquitin proteasome and autophagy pathways, inhibiting the host’ s gene silencing [38]. In this study, treatment with MG132 or 3MA confirmed that NbMLP43 was degraded via the UPS pathway. In the process of co-evolution with their hosts, pathogenic agents have evolved mechanisms to inhibit and/or use the UPS to promote self-infection and replication. For example, the newcastle disease virus V protein recruits E3 ubiquitin ligase RNF5 for polyubiquitination and degrades the mitochondrial antiviral signal protein, thereby inhibiting interferon production and facilitating self-infection [39]. Human immunodeficiency virus (HIV)-1 Nef promotes the degradation of p53 protein and protects HIV-1-infected cells from p53-induced apoptosis [35]. These findings provide important insights into the principles of innate immunity. We investigated NbMLP43, as the target of the UPS, given its correlation with viral infections and used PVY to infect NbMLP43 (K38R) and NbMLP43-OE plants. We found that NbMLP43 ubiquitination decreased after mutation in the K38 site, that NbMLP43 expression was upregulated, and that PVY CP expression was downregulated. These results suggest that the ubiquitinated degradation of NbMLP43 contributed to viral infection.

The B-box (BBX) family is a subfamily of zinc finger proteins. Compared with those on animal BBX protein functions [46,47,48,49], studies on the function of BBX proteins in plants are relatively limited. However, recent studies have found that plant BBX domains play an important role in mediating protein interactions and regulating gene expression [50,51,52]. In this study, we identified the interacting protein NbBBX24 in NbMLP43, a light responsive factor that upregulated the expression of *NbBBX24* and that of the infecting virus. Moreover, NbMLP43 ubiquitination was decreased after silencing *NbBBX24*, whereas NbMLP43 expression was upregulated, and that of PVY CP was downregulated. Given that *Arabidopsis* STO/BBX24 negatively regulates UV-B-mediated stress by interacting with COP1 and inhibiting HY5 transcriptional activity [53], we postulate that NbBBX24 is involved in the ubiquitination and degradation of NbMLP43 as an important intermediate component that promotes viral infection.

In the arms race between host immunity and virus anti-immunity, we identified a new target of the UPS using PVY and *N. benthamiana* as a model research host system. We identified that NbMLP43 displays broad resistance to plant RNA viruses, is specifically bound by NbBBX24, and is subsequently degraded. This work enriches theoretical studies on the role of the UPS in the innate immunity of virus–host interactions. Notably, our results showed that non-ubiquitinated NbMLP43(K38R) conferred stronger resistance to several RNA viruses and provided a basis for the subsequent utilization of this resistant protein (Figure 7). However, no specific E3 ligase-targeting NbMLP43 was identified, and whether the light signal factor NbBBX24 positively regulates viral infection based on NbMLP43 ubiquitination and degradation requires further research.

## Figures and Tables

**Figure 1 cells-12-00590-f001:**
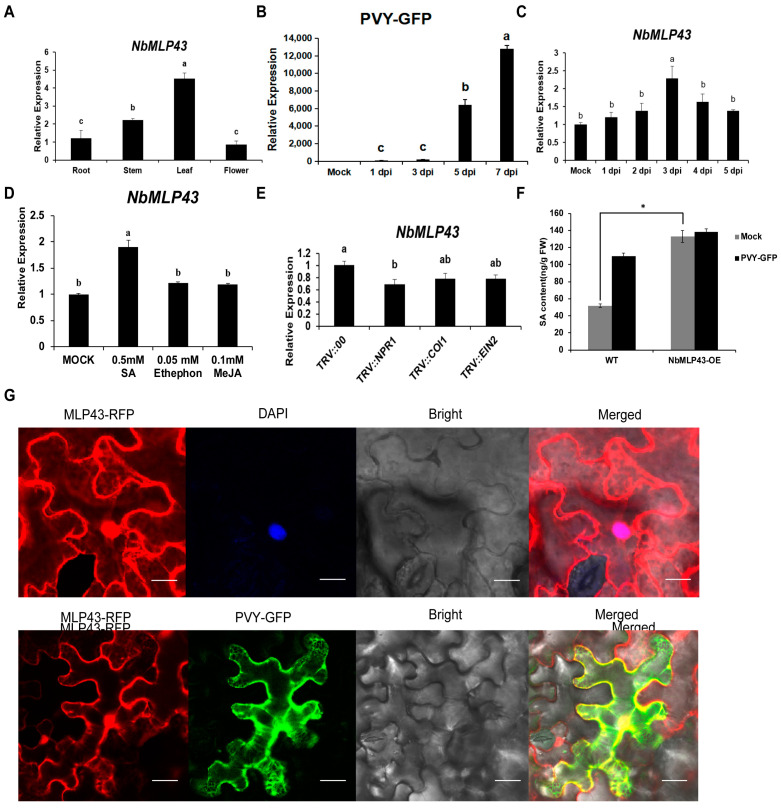
Expression pattern analysis of *NbMLP43*. (**A**) Trends in *NbMLP43* expression in the root, stem, leaf, and flower. Data were analyzed with Duncan’s multiple range tests; different letters indicate that the values of the four treatments were significantly different (*p* < 0.05), which were analyzed the same as figures (**B**–**E**). (**B**) Trends in PVY-GFP trend at 1, 3, 5, 7 dpi. (**C**) Trends in *NbMLP43* expression after PVY inoculation. (**D**) *NbMLP43* expression after spraying with 0.5 mM SA, 0.05 mM ethephon, or 0.1 mM Me-JA; water with 0.02% Tween 20 was used as a control. (**E**) *NbMLP43* expression after silencing key signaling genes, namely *NPR1*, *COI1*, and *EIN2*. (**F**) SA content was detected before and after PVY infection in NbMLP43-OE plants, with wild-type as the control. Data were analyzed with independent sample *t*-test; * indicated that values of the two treatments were significantly different (*p* < 0.05). (**G**) Subcellular distribution of NbMLP43 in plant’s epidermis before and after PVY-infection.

**Figure 2 cells-12-00590-f002:**
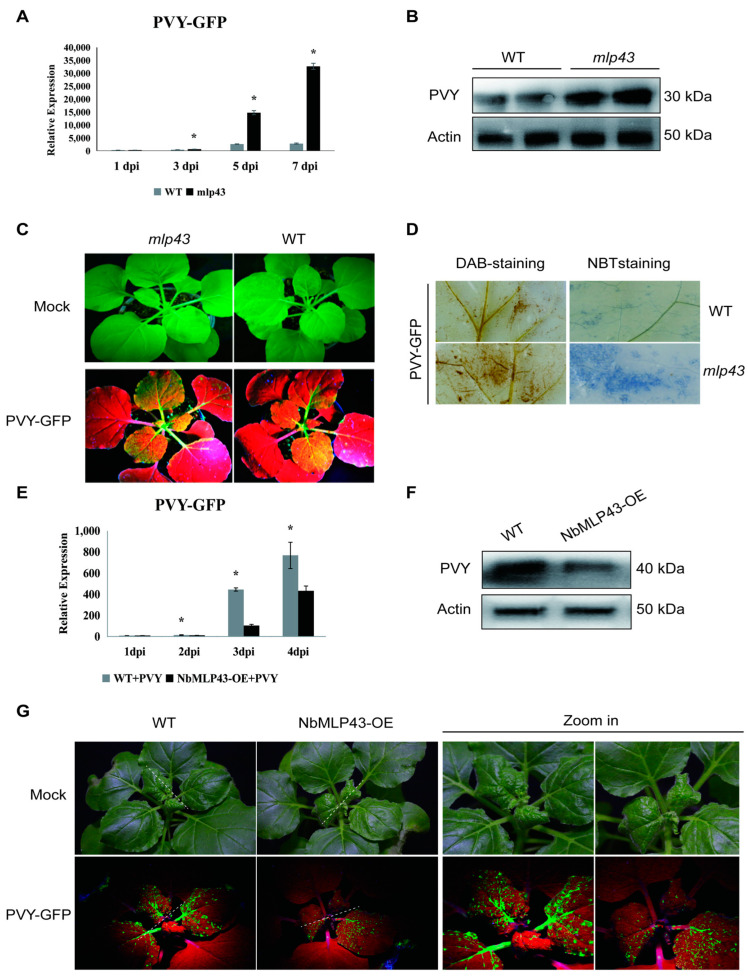
Functional validation of NbMLP43 based on *mlp43* and NbMLP43-OE plants. (**A**) Detection of PVY CP expression in *mlp43* and WT at 1, 3, 5, and 7 dpi at the RNA level. Data were analyzed with independent sample *t*-test; * indicated that values of the two treatments were significantly different (*p* < 0.05), which were analyzed the same way as figure E. (**B**) PVY CP protein levels in *mlp43* and WT at 5 dpi. (**C**) PVY-GFP fluorescence in *mlp43* and WT, with mock as a negative control. (**D**) DAB and NBT staining of WT and *mlp43* plants inoculated with PVY-GFP. (**E**) Real-time PCR was used to detect the differences in virus expression at 1, 2, 3, and 4 dpi. NbMLP43-OE/PVY was the treatment group, whereas WT/PVY was the control group. (**F**) Differential expression of PVY CP protein was detected at 4 dpi. (**G**) GFP fluorescence at 12 dpi.

**Figure 3 cells-12-00590-f003:**
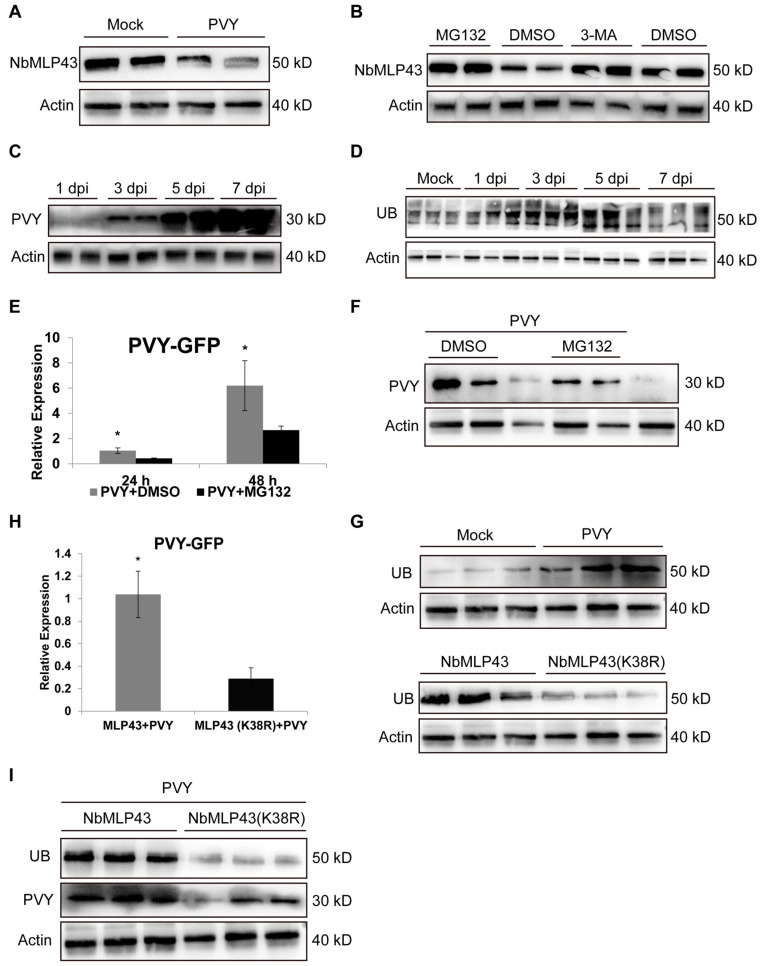
NbMLP43 was degraded through the UPS pathway. (**A**) PVY infection decreased NbMLP43 expression at the protein level. (**B**) NbMLP43 expression was detected after MG132 and 3MA treatment, with DMSO as a control. (**C**) Detection of PVY CP at 1, 3, 5, and 7 dpi. (**D**) Ubiquitination level in PVY infected plants at 1, 3, 5, and 7 dpi, with PBS treatment as mock. (**E**) PVY CP was detected at the RNA level after treatment with MG132 at 24 h and 48 h, and DMSO was treated as a control. Data were analyzed with independent sample *t*-test; * indicated that values of the two treatments were significantly different (*p* < 0.05), which were analyzed the same way as figure H. (**F**) PVY CP protein level after treatment with MG132 at 48 h. (**G**) Ubiquitination level of NbMLP43 after PVY infection using ubiquitin antibody and after K38 mutation. (**H**) Effect of NbMLP43 ubiquitination on PVY infection at the RNA level. (**I**) Effect of NbMLP43 ubiquitination on PVY infection at the protein level.

**Figure 4 cells-12-00590-f004:**
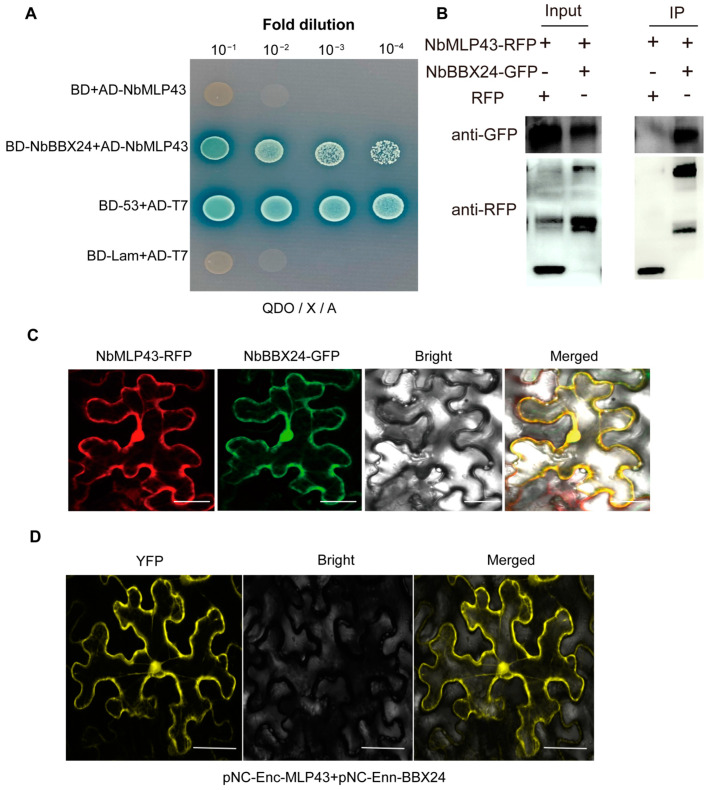
NbBBX24 interacted with NbMLP43. (**A**) Yeast point-to-point validation of the interaction between NbMLP43 and NbBBX24. (**B**) Interaction between NbMLP43 and NbBBX24 was verified with co-IP assay. (**C**) Subcellular distribution of NbMLP43 and NbBBX24. (**D**) YFP interaction signal of NbMLP43 and NbBBX24.

**Figure 5 cells-12-00590-f005:**
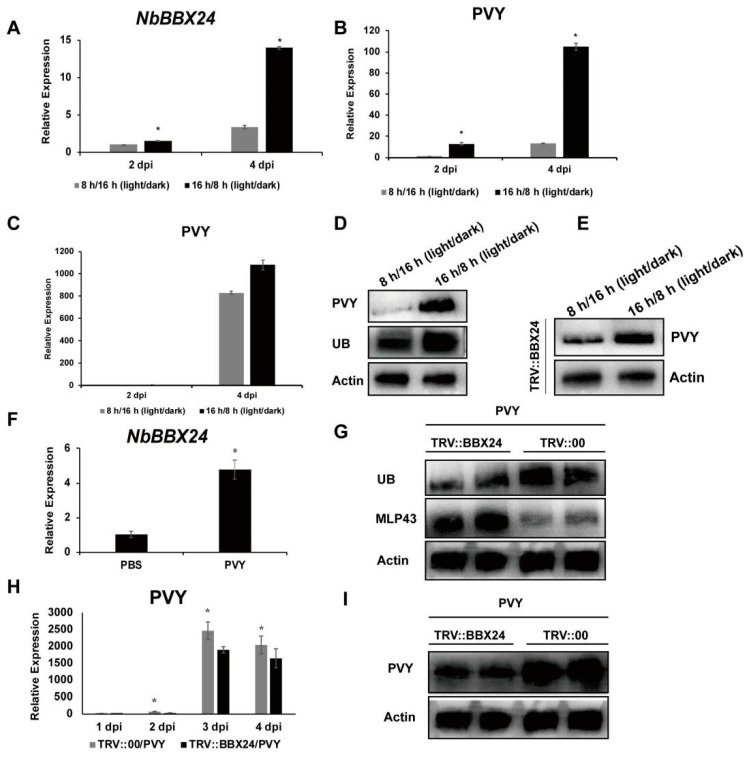
NbBBX24 acted as an intermediate factor that mediated the ubiquitinated degradation of NbMLP43. (**A**) Transcriptional levels of NbBBX24 were detected with light/dark photoperiods of 16/8 h and 8/16 h at 2 and 4 dpi under PVY infection. Data were analyzed with independent sample *t*-test; * indicated that values of the two treatments were significantly different (*p* < 0.05), the same as figure (**B**,**C**,**F**,**H**). (**B**) Transcriptional levels of PVY CP were detected with light/dark photoperiods of 16-h/8-h and 8-h/16-h at 2 and 4 dpi. (**C**) After silencing NbBBX24, transcriptional levels of PVY CP at light/dark photoperiods of 16/8 h and 8/16 h at 2 and 4 dpi under PVY infection. (**D**) Protein levels of PVY CP and ubiquitination levels of NbMLP43 at light/dark photoperiods of 16/8 h and 8/16 h at 4 dpi. (**E**) After silencing *NbBBX24*, protein levels of PVY CP at light/dark photoperiods of 16/8 h and 8/16 h at 4 dpi. (**F**) Expression of *NbBBX24* under PVY infection at the transcriptional level. (**H**) Effects of silencing *NbBBX24* on PVY infection at the transcriptional level. (**G**,**I**) Effects of silencing *NbBBX24* on NbMLP43 ubiquitination, NbMLP43 itself, and PVY infection at the protein level.

**Figure 6 cells-12-00590-f006:**
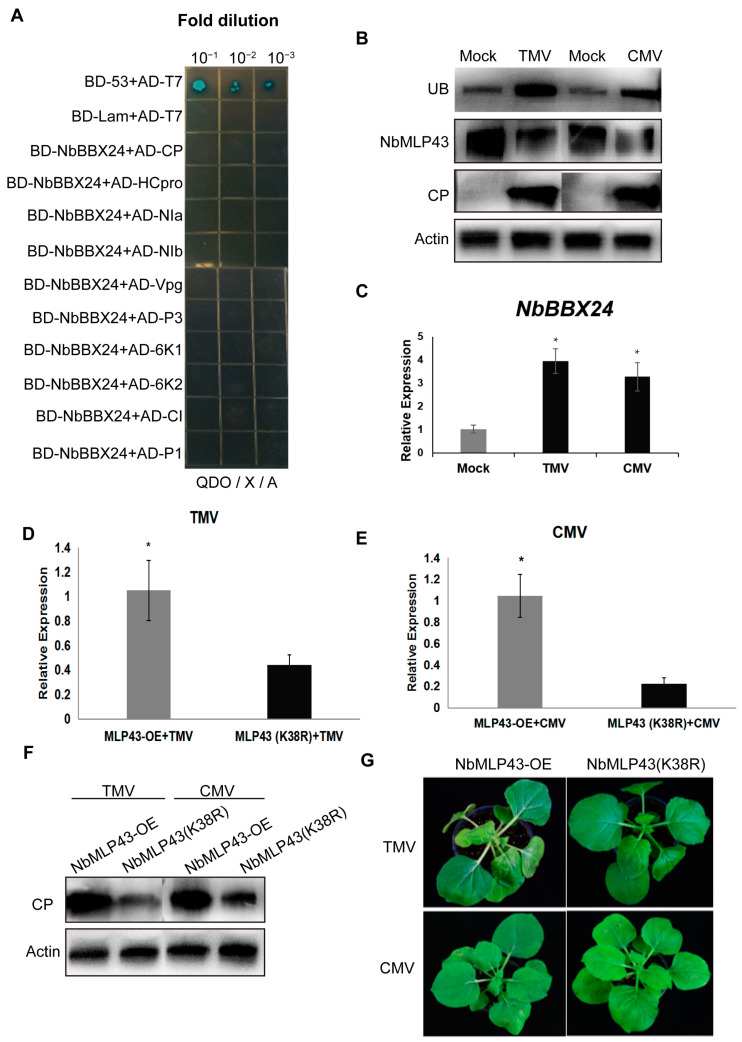
Viral infection nonspecifically promoted ubiquitination of NbMLP43. (**A**) Yeast point-to-point validation of the interaction between NbBBX24 and PVY. (**B**) Effects of TMV (or CMV) infection on NbMLP43 ubiquitination and NbMLP43 itself. (**C**) Effects of TMV and CMV infection on the expression of NbBBX24 at the transcriptional level. Data were analyzed with independent sample *t*-test; * indicated that values of the two treatments were significantly different (*p* < 0.05), the same as figures (**D**,**E**). (**D**) Effect of NbMLP43 ubiquitination on TMV infection at the transcriptional level. (**E**) Effect of NbMLP43 ubiquitination on CMV infection at the transcriptional level. (**F**) Effects of NbMLP43 ubiquitination on TMV and CMV infection at the protein level. (**G**) Effects of NbMLP43 ubiquitination on TMV and CMV infection as identified from plant disease symptoms.

**Figure 7 cells-12-00590-f007:**
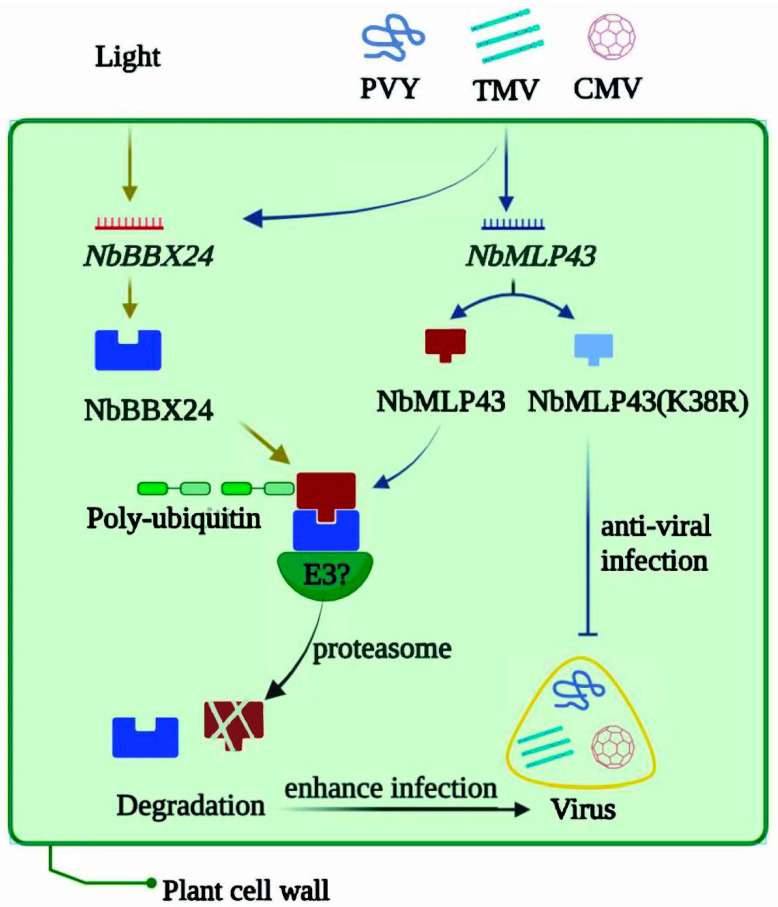
Article overview of virus hijacked UPS to degrade resistance protein NbMLP43. Viral infection induced NbMLP43 transcription but decreased NbMLP43 at the protein level. NbMLP43 conferred resistance to *N. benthamiana* against viral infection. The light response factor NbBBX24 specifically binded NbMLP43 for its ubiquitination and degradation via the UPS, enhancing viral infection. And non-ubiquitinated NbMLP43(K38R) conferred stronger resistance to RNA viruses.

**Table 1 cells-12-00590-t001:** Analysis of the cis-acting regulatory element of the *NbMLP43* promoter.

Number	Site Name	Amount	Sequence	Function of Site
1	GARE-motif	1	TCTGTTG	*Brassica* oleraceagibberellin-responsive element
2	I-box	1	GTATAAGGCC	Part of a light responsive element
3	Box 4	3	ATTAAT	Part of a conserved DNA module involved in light responsiveness
4	TC-rich repeats	1	GTTTTCTTAC	Cis-acting element involved in defense and stress responsiveness
5	CAAT-box	45	CCAAT	Cis-acting element in promoter and enhancer regions
6	AT-rich element	2	ATAGAAATCAA	Binding site of AT-rich DNA binding protein (ATBP-1)
7	G-Box	2	CACGTT	Cis-acting regulatory element involved in light responsiveness
8	O2-site	1	GATGA(C/T)(A/G)TG(A/G)	Cis-acting regulatory element involved in zein metabolism regulation
9	G-box	3	TAAACGTG	Cis-acting regulatory element involved in light responsiveness
10	GT1-motif	1	GGTTAA	Part of a light responsive element
11	MRE	1	AACCTAA	Binding site involved in light responsiveness
12	TATA-box	64	TATA	Core promoter element around -30 of transcription start
13	ARE	8	AAACCA	Cis-acting regulatory element essential for the anaerobic induction
14	LTR	2	CCGAAA	Cis-acting element involved in low-temperature responsiveness
15	TCA-element	1	TCAGAAGAGG	Cis-acting element involved in salicylic acid responsiveness
16	TGA-element	2	AACGAC	Auxin-responsive element
17	TCT-motif	1	TCTTAC	Part of a light responsive element
18	ABRE	4	ACGTG	Cis-acting element involved in the abscisic acid responsiveness

## Data Availability

The-full length sequence of NbMLP43 is available in GenBank under the accession number: MK780770.

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
