# Peer review of "NbMLP43 Ubiquitination and Proteasomal Degradation via the Light Responsive Factor NbBBX24 to Promote Viral Infection"

_cells, 2023, doi:10.3390/cells12040590_

Round 1

Reviewer 1 Report

The work described in this manuscript is quite novel and very interesting to the readership of “cells”. While many of the conclusions are justified by the data, others are not, due to the absence of appropriate controls (see comments 37, 38, 47-52) or lack of description (comments 28, 29, 31, 32, 35, 36, 41, 45, 46). Although the work in the Results and Discussion sections are quite well written, the Introduction (when describing the work of others) and the Materials and Method sections (describing how the work was done) were not as carefully crafted and need careful editing (comments 1-27, 30, 33, 34, 39, 40, 42-44 and 53-61). While the last can be easily corrected, the first issue requires additional data be presented (in the figures of the text or in supplementary figures) before the conclusions can be substantiated properly.

Another issue relates to the complexity of the MPDI submission process, which is not always readily understandable to authors whose native language is not English. Here, the authors did not understand the difference between three sets of data that may be needed (additional files not to be published – this is rather cryptic and presumably means either large datasets, or data intended for publication in another work, being submitted to substantiate a point being examined or discussed), or are required – both supplementary files and unprocessed images (such as the original gel photos or blot images, from which regions had been cut and pasted into a different image). The last is required to verify the absence of either image manipulation, or presence of multiple bands that can impact upon the specificity claimed. Here, the authors basically sent all images three  times: once for the paper and supplementary files; once as additional files; and once as unprocessed images (in place of full-size gels and blots). Hence, we do not have available the unprocessed data for Figures 1G, 2B, 2D, 2F, 3A-D, 3F, 3G, 3I, 4B, 4C, 5D, 5E, 5G, 5I, 6B, and 6F. Thus, the submitted materials were not in a form that can be accepted at this time.

Specific comments:

1. ln 20. What do the authors mean by “self-infection”? This is not a standard term.

2. ln 24. Change “enrich” to “enriches”.

3. ln 25-26. Keywords should all be in the same font size.

4. ln 29. In other MDPI journals, when two consecutive references are cited, they are separated by a comma, rather than a dash, which is used for three of more consecutive references. See also ln 43, 46, 47, 53, 414-415, and 426.

5. ln 31. Change “Particularly,” to “In particular,”.

6. ln 35. Change “increasing” to “further”.

7. ln 44. “MLP gene”. When abbreviations of terms refer to genes, the terms should be in italics. See also ln 45, 47, 154, and 436.

8. ln 45. Are the authors referring to transcription of both MLP28 and MLP3? If so, then the sentence should be rephrased to “Transcription in Arabidopsis thaliana of MLP28 (AT1G70830) and MLP3 were induced by………, respectively [13,14].”

9. ln 47-48. There is no need to re-define MLP for differently numbered members of this protein/gene family, especially, in the second use of the term (ln 45 vs. ln 47-48). The same for MLP43 on ln 49-50. See also ln 93, 205, 209, 211, 212, 214, and 416.

10. ln 50. Change “whose expression is upregulated in the mRNA levels…” to “the mRNA expression of which is upregulated…”. [Who/whom and whose are only used for people and not animals of things.]

11. ln 53. Rephrase. It reads now as if PVY can be infected. Change to “Infection by PVY causes significant harm to commercial crops [18,19],”.

12. ln 54. Change “were” to “was” and “materials” to “host system”.

13. ln 55. Change “combing” to “combined”.

14. ln 56. Change “with” to “by”.

15. ln 69. Change “materials” to “plants”.

16. ln 70-71. Rephrase to “The following viruses were used in these analyses: CMV, TMV, PVY, and PVY expressing the green fluorescent protein (PVY-GFP) [20].”

17. ln 73-74. Rephrase to “Two expression vectors were used, each harboring a fusion gene encoding the red fluorescent protein (RFP) tag: p35S::MLP43-RFP and p35S::MLP43(K38R)-RFP.”

18. ln 79-80. Rephrase to “The p35S::BBX24-GFP expression vector was constructed by homologous recombination…….”.

19. ln 83. Change “base” to “gene”.

20. ln 84. Insert “corresponding” before “CDS”. The same on ln 88.

21. ln 90. You do not have to say the tissue was weighed if you give the weight.

22. ln 92. Italicize “N. benthamiana”. See also ln 154, 341-342, 484, 485, and 487 (and correct Benthamiana to benthamiana on ln 487).

23. ln 95. Insert “and the sequence was“ before “submitted”.

24. ln 98. Change “amplifythe” to “amplify” and ‘the”.

25. ln 104-105. Insert “based on the tobacco rattle virus (TRV) VIGS system,” after “assays”, and change “Agrobacterium” to “Agrobacterium tumefaciens”; the latter also on ln 177.

26. ln 107. Change to read “The gene silencing efficiency was determined after 14 days.”

27. ln 108. Change to read “To determine the effects of VIGS on the viral infection, PVY….selected time points to detect…..CP.”

28. ln 113-115. More details or references are needed here to describe the preparation of N. benthamiana protoplasts.

29. ln 123. What do the authors mean by “biological replicates”? Do they mean separate samples from the same leaf, separate plants done in the same experiment, or separate plants done in completely separate experiments, at different times? While the middle one constitutes a “biological replicate”, only the latter one constitutes a true independent repeat. See also ln 140-141, ln 146 and ln 157. The statistical analysis (ln 191-195) indicates that the experiments were independent rather than just biological replicates.

30. ln 125. The authors need to give the name of the manufacturer.

31. ln 138-139. How was the SA content determined? This is not mentioned.

32. ln 145. The “backs of the leaves” is not scientifically clear. Do the authors mean the lower surface (the abaxial surface), or the upper surface (the adaxial surface)? See also ln 180.

33. ln 163. Insert “in a shaker incubator” after “24 h”.

34. ln 198. Change to “PVY is a member of the genus Potyvirus….”. Virus species are conceptual constructs and not real things.  Real viruses can infect plants, while conceptual constructs cannot. Therefore, taxonomically, PVY is not a species, since the ICTV has ruled that abbreviations should not be used to define taxonomic conceptual constructs, but only real viruses. Similarly, real virus names are written in Roman letters while conceptual construct names (species, genera, families, etc.) are written in italics.

35. ln 200-206. There was no description in the Materials and Methods section concerning the preparation of RNAs for RNA-seq nor the methods used for transcriptional analysis. This information needs to be given.

36. ln 215 and 426. Where is the Bet v 1 domain located in MbMLP43?

37. ln 225-226. The authors cannot state that the virus accumulation peaked at 7 dpi, since they took no samples later than 7 dpi.

38. ln 228 and Fig. 1G. If something is dispersed through the cytoplasm and also in the nucleus, it is not referred to as “localized”. That term usually means it is in one place only. Rather, the term “distributed” is used instead. The same for section 2.5 and Fig. S4.

39. In Fig. S4, what are the scales of the images? Scale bars should be added, since it looks like the magnification of PVY-GFP and DAPI were much lower than for the first two samples.

40. ln 234. Insert “, respectively,” after “(ET)”.

41. ln 256 vs. Fig. 2A and the legend to Fig. 2A. The text and the figure show sampling at days 1, 3, 5, and 7, while the figure legend states the sampling was at days 1, 2, 4, and 5. Similarly, the text says that the CP levels show in Fig. 2B were at 5 dpi, while the figure legend says 4 dpi. These need to be consistent, and correct.

42. ln 290. Insert “(Fig. 1C)” after “transcription”, since Fig. 3A only shows the results of protein accumulation.

43. ln 293. Change to “NbMLP43 levels were higher in MG132-treated plants than in control (DMSO) plants, whereas….”.

44. ln 300. Change Fig. 3C to Fig. 3D. [Fig. 3C is the PVY CP levels.] Also. The levels were highest at 5 dpi, not 4 dpi (which was not measured).

45. ln 309-312. Data not shown?

46. ln 336. A proper figure legend is needed for Fig. S6.

47. ln 338-342. Why were neither the reciprocal interaction (in BD vs. AT plasmids) nor reciprocal single protein/single empty plasmid combinations done to eliminate (a) artifactual interaction and self-activation of one protein alone? These are standard controls for Y2H experiments.

48. ln 342-343. Fig. 4C does not show they colocalize, but simply that they are both present in the both the nucleus and cytoplasm.

49. ln 343-344. Fig. 4B has no markers and no positive controls to clearly identify what the various bands are. Is the RFP a mutant of GFP, or is this dsRed (RFP); i.e., from a different species?

50. ln 350. From the information supplied, none of the three methods used have sufficient controls to verify interaction.

51. ln 379. The lack of an interaction between PVY encoded proteins and NbBBX24 may be correct, although the lack of reciprocal interactions and tests for stability of the PVY proteins expressed in this Y2H system leads to an equivocal conclusion. There is also another potyvirus protein designated P3N-PIPO, produced by translational slippage and early termination in the P3 ORF, although I am not sure that it has been identified in PVY.

52. ln 382. No data were shown in Figure 6 for the overexpressing NbMLP43 plants, and the “respectively” should be deleted. Similarly, on ln 402, there is no validation of any interaction between PVY encoded proteins and NbBBX24. Rather, there is an absence of interaction detected between these proteins.

53. The Materials and Methods section should mention TMV and CMV for those methods used in Figure 6 concerning these two viruses. Only PVY is mentioned in the various methods.

54. ln 393 and 395. Neither situation is “respectively”. Accumulation was detected under both light regimes and at both times.

55. ln 407. Change “Fffect” to “Effect”.

56. ln 417-418. You do not need both “induced” and “upregulated”; “induced” is preferable here.

57. ln 422. Change “guidance” to “guide”.

58. ln 430. Rephrase. The MLP28 does not resist virus infection, since a protein cannot be infected. MLP28 may help the host plant resist virus infection.

59. ln 443. Change “achieve” to “promote”. The virus achieves self-infection once it can replicate and move. Moreover, replication is achieved through other processes than the ones described here. The processes described in this work relate to promoting self-infection by inhibiting factors that are involved in the prevention of virus infection.

60. ln 470. Change “and subsequently degrades” to “and is subsequently degraded”.

61. ln 473. Insert “several” before “RNA viruses”. There may be member of some RNA virus families that are able to prevent this response.

Author Response

Response to Reviewer 1 Comments

Point 1: ln 20. What do the authors mean by “self-infection”? This is not a standard term. 

Response 1: Thank you for your useful comments. “viral self-infection” has been changed to “viral infection”.

Point 2: ln 24. Change “enrich” to “enriches”.

Response 2: Thank you for your useful comments. The change has been made accordingly.

Point 3: ln 25-26. Keywords should all be in the same font size.

Response 3: Thank you. The change has been made accordingly.

Point 4: ln 29. In other MDPI journals, when two consecutive references are cited, they are separated by a comma, rather than a dash, which is used for three of more consecutive references. See also ln 43, 46, 47, 53, 414-415, and 426.

Response 4: Thank you for your suggestion. The changes have been made accordingly.

Point 5: ln 31. Change “Particularly,” to “In particular,”.

Response 5: Thank you. The change has been made accordingly.

Point 6: ln 35. Change “increasing” to “further”.

Response 6: Thank you. As suggested, the change has been made accordingly.

Point 7: ln 44. “MLP gene”. When abbreviations of terms refer to genes, the terms should be in italics. See also ln 45, 47, 154, and 436.

Response 7: Thank you. As suggested, the changes have been made accordingly.

Point 8: ln 45. Are the authors referring to transcription of both MLP28 and MLP3? If so, then the sentence should be rephrased to “Transcription in Arabidopsis thaliana of MLP28 (AT1G70830) and MLP3 were induced by………, respectively [13,14].”

Response 8: Thank you. As suggested, the change has been made accordingly.

Point 9: ln 47-48. There is no need to re-define MLP for differently numbered members of this protein/gene family, especially, in the second use of the term (ln 45 vs. ln 47-48). The same for MLP43 on ln 49-50. See also ln 93, 205, 209, 211, 212, 214, and 416.

Response 9: Thank you for your suggestion. The changes have been made accordingly.

Point 10: ln 50. Change “whose expression is upregulated in the mRNA levels…” to “the mRNA expression of which is upregulated…”. [Who/whom and whose are only used for people and not animals of things.]

Response 10: Thank you for your useful comments. The change has been made accordingly.

Point 11: ln 53. Rephrase. It reads now as if PVY can be infected. Change to “Infection by PVY causes significant harm to commercial crops [18,19],”.

Response 11: Thank you. As suggested, the change has been made accordingly.

Point 12: ln 54. Change “were” to “was” and “materials” to “host system”.

Response 12: Thank you. As suggested, we have changed “were” to “was” and “materials” to “host system”.

Point 13: ln 55. Change “combing” to “combined”.

Response 13: Thank you. As suggested, we have changed “combing” to “combined”.

Point 14: ln 56. Change “with” to “by”.

Response 14: Thank you. As suggested, we have changed “with” to “by”.

Point 15: ln 69. Change “materials” to “plants”.

Response 15: Thank you. As suggested, we have changed “materials” to “plants”.

Point 16: ln 70-71. Rephrase to “The following viruses were used in these analyses: CMV, TMV, PVY, and PVY expressing the green fluorescent protein (PVY-GFP) [20].”

Response 16: Thank you for your suggestion. The change has been made accordingly.

Point 17: ln 73-74. Rephrase to “Two expression vectors were used, each harboring a fusion gene encoding the red fluorescent protein (RFP) tag: p35S::MLP43-RFP and p35S::MLP43(K38R)-RFP.”

Response 17: Thank you for your suggestion. The change has been made accordingly.

Point 18: ln 79-80. Rephrase to “The p35S::BBX24-GFP expression vector was constructed by homologous recombination…….”.

Response 18: Thank you. As suggested, the change has been made accordingly.

Point 19: ln 83. Change “base” to “gene”.

Response 19: Thank you for your suggestion. “base”has been changed to “gene”.

Point 20: ln 84. Insert “corresponding” before “CDS”. The same on ln 88.

Response 20: Thank you for your suggestion. The changes have been made accordingly.

Point 21: ln 90. You do not have to say the tissue was weighed if you give the weight.

Response 21: Thank you. As suggested, “weighed” has been changed to “prepared”.

Point 22: ln 92. Italicize “N. benthamiana”. See also ln 154, 341-342, 484, 485, and 487 (and correct Benthamiana to benthamiana on ln 487).

Response 22: Thank you for your useful comments. The changes have been made accordingly.

Point 23: ln 95. Insert “and the sequence was“ before “submitted”.

Response 23: Thank you for your suggestion. “the sequence was” has been inserted.

Point 24: ln 98. Change “amplifythe” to “amplify” and ‘the”.

Response 24: Thank you for your useful comments. The change has been made accordingly.

Point 25: ln 104-105. Insert “based on the tobacco rattle virus (TRV) VIGS system,” after “assays”, and change “Agrobacterium” to “Agrobacterium tumefaciens”; the latter also on ln 177.

Response 25: Thank you for your useful comments. The changes have been made accordingly.

Point 26: ln 107. Change to read “The gene silencing efficiency was determined after 14 days.”

Response 26: Thank you for your useful comments. The change has been made accordingly.

Point 27: ln 108. Change to read “To determine the effects of VIGS on the viral infection, PVY….selected time points to detect…..CP.”

Response 27: Thank you for your useful comments. The changes have been made accordingly.

Point 28: ln 113-115. More details or references are needed here to describe the preparation of N. benthamiana protoplasts.

Response 28: Thank you for your suggestion. More details and reference have been added.

Point 29: ln 123. What do the authors mean by “biological replicates”? Do they mean separate samples from the same leaf, separate plants done in the same experiment, or separate plants done in completely separate experiments, at different times? While the middle one constitutes a “biological replicate”, only the latter one constitutes a true independent repeat. See also ln 140-141, ln 146 and ln 157. The statistical analysis (ln 191-195) indicates that the experiments were independent rather than just biological replicates.

Response 29: Thank you for your useful comments. Each experiments were repeated three times ( separate plants done in completely separate experiments), with each experiment containing three biological replicates. As suggested, The changes have been made accordingly.

Point 30: ln 125. The authors need to give the name of the manufacturer.

Response 30: Thank you. As suggested, the manufacturer has been added.

Point 31: ln 138-139. How was the SA content determined? This is not mentioned.

Response 31: Thank you for your suggestion. The method of quantitative detection of SA has been added.

Point 32: ln 145. The “backs of the leaves” is not scientifically clear. Do the authors mean the lower surface (the abaxial surface), or the upper surface (the adaxial surface)? See also ln 180.

Response 32: Thank you for your useful comments. We mean the lower surface (the abaxial surface), and the changes have been made accordingly.

Point 33: ln 163. Insert “in a shaker incubator” after “24 h”.

Response 33: Thank you for your suggestion. “in a shaker incubator” has been inserted.

Point 34: ln 198. Change to “PVY is a member of the genus Potyvirus….”. Virus species are conceptual constructs and not real things.  Real viruses can infect plants, while conceptual constructs cannot. Therefore, taxonomically, PVY is not a species, since the ICTV has ruled that abbreviations should not be used to define taxonomic conceptual constructs, but only real viruses. Similarly, real virus names are written in Roman letters while conceptual construct names (species, genera, families, etc.) are written in italics.

Response 34: Thank you for your useful comments. The change has been made accordingly.

Point 35: ln 200-206. There was no description in the Materials and Methods section concerning the preparation of RNAs for RNA-seq nor the methods used for transcriptional analysis. This information needs to be given.

Response 35: Thank you for your suggestion. We have added the description in the Materials and Methods section concerning the preparation of RNAs for RNA-seq and the methods used for transcriptional analysis.

Point 36: ln 215 and 426. Where is the Bet v 1 domain located in MbMLP43?

Response 36: Thank you for your useful comments. The Bet v 1 domain located in MbMLP43 is 2-146 aa, and we have added in the manuscript.

Point 37: ln 225-226. The authors cannot state that the virus accumulation peaked at 7 dpi, since they took no samples later than 7 dpi.

Response 37: Thank you for your useful comments. “peaking at 7 dpi” has been deleted.

Point 38: ln 228 and Fig. 1G. If something is dispersed through the cytoplasm and also in the nucleus, it is not referred to as “localized”. That term usually means it is in one place only. Rather, the term “distributed” is used instead. The same for section 2.5 and Fig. S4.

Response 38: Thank you for your useful comments. The changes have been made accordingly.

Point 39: In Fig. S4, what are the scales of the images? Scale bars should be added, since it looks like the magnification of PVY-GFP and DAPI were much lower than for the first two samples.

Response 39: Thank you for your useful comments. The changes have been made accordingly.

Point 40: ln 234. Insert “, respectively,” after “(ET)”.

Response 40: Thank you for your suggestion. The change has been made accordingly.

Point 41: ln 256 vs. Fig. 2A and the legend to Fig. 2A. The text and the figure show sampling at days 1, 3, 5, and 7, while the figure legend states the sampling was at days 1, 2, 4, and 5. Similarly, the text says that the CP levels show in Fig. 2B were at 5 dpi, while the figure legend says 4 dpi. These need to be consistent, and correct.

Response 41: Thank you for your useful comments. We have corrected the legend of Fig. 2A, which was show sampling at days 1, 3, 5, and 7. And Fig. 2B legend was samples at 5 dpi.

Point 42: ln 290. Insert “(Fig. 1C)” after “transcription”, since Fig. 3A only shows the results of protein accumulation.

Response 42: Thank you for your suggestion. We have inserted “(Fig. 1C)” after “transcription”.

Point 43: ln 293. Change to “NbMLP43 levels were higher in MG132-treated plants than in control (DMSO) plants, whereas….”.

Response 43: Thank you for your suggestion. The change has been made accordingly.

Point 44: ln 300. Change Fig. 3C to Fig. 3D. [Fig. 3C is the PVY CP levels.] Also. The levels were highest at 5 dpi, not 4 dpi (which was not measured).

Response 44: Thank you for your useful comments. The changes have been made accordingly.

Point 45: ln 309-312. Data not shown?

Response 45: Thank you for your useful comments. We have supplied the data in Supplementary Table S2.

Point 46: ln 336. A proper figure legend is needed for Fig. S6.

Response 46: Thank you for your suggestion. The figure legend of Fig. S6 has been revised to “Screening of interactive proteins for NbMLP43 in yeast two-hybrid assay.”.

Point 47: ln 338-342. Why were neither the reciprocal interaction (in BD vs. AT plasmids) nor reciprocal single protein/single empty plasmid combinations done to eliminate (a) artifactual interaction and self-activation of one protein alone? These are standard controls for Y2H experiments.

Response 47: Thank you for your suggestion. Fig. 4A has been revised. Self-activation assays of BD-NbBBX24 has been performed in Fig. S7.

Point 48: ln 342-343. Fig. 4C does not show they colocalize, but simply that they are both present in the both the nucleus and cytoplasm.

Response 48: Thank you for your useful suggestion. The description of Fig. 4C was changed basing on your comments.

Point 49: ln 343-344. Fig. 4B has no markers and no positive controls to clearly identify what the various bands are. Is the RFP a mutant of GFP, or is this dsRed (RFP); i.e., from a different species?

Response 49: So sorry for the inconvenience caused to your work, we have revised Fig. 4B. Protein samples co-expressed with NbMLP43 and NbBBX24 were blotted with anti-GFP or anti-RFP antibodies, serving RFP as negative control. The results showed that NbMLP43, NbBBX24 and RFP were all expressed (Fig. 4B Input). The total protein was purified using magnetic beads carrying an anti-RFP antibody, and the eluted protein samples were RFP-containing proteins and plant endogenous proteins interacting with NbMLP43, the results showed that NbBBX24 was only precipitated in the presence of NbMLP43 (Fig. 4B IP). These results further confirmed the interaction between NbBBX24 and NbMLP43.

Point 50: ln 350. From the information supplied, none of the three methods used have sufficient controls to verify interaction.

Response 50: So sorry for the inconvenience caused to your work, we have re-layouted Fig. 4 to  verify interaction. In addition, we used BIFC to verify the in vivo interaction between NbMLP43 and NbBBX24, YFP signal was detected in leaves co-infiltrated with pNC-Enc-MLP43 and pNC-Enn-BBX24 (Fig. 4D).

Point 51: ln 379. The lack of an interaction between PVY encoded proteins and NbBBX24 may be correct, although the lack of reciprocal interactions and tests for stability of the PVY proteins expressed in this Y2H system leads to an equivocal conclusion. There is also another potyvirus protein designated P3N-PIPO, produced by translational slippage and early termination in the P3 ORF, although I am not sure that it has been identified in PVY.

Response 51: Thanks for your helpful advice. In this experiment, we considered direct protein interactions and did not perform in vivo interaction. The manuscript has been revised to “There was no directly interaction between PVY encoded proteins and NbBBX24”. Cross-validation of P3N-PIPO was not performed in this experiment. Thank you for the good suggestion, we can perform a comprehensive validation of the protein interaction in the follow-up experiments.

Point 52: ln 382. No data were shown in Figure 6 for the overexpressing NbMLP43 plants, and the “respectively” should be deleted. Similarly, on ln 402, there is no validation of any interaction between PVY encoded proteins and NbBBX24. Rather, there is an absence of interaction detected between these proteins.

Response 52: Thank you for your suggestion. The change has been made accordingly, and Fig. 6D-G has been re-edited, displaying NbMLP43-OE.

Point 53: The Materials and Methods section should mention TMV and CMV for those methods used in Figure 6 concerning these two viruses. Only PVY is mentioned in the various methods.

Response 53: Thank you for your suggestion. We have added methods concerning TMV and CMV in the Materials and Methods section.

Point 54: ln 393 and 395. Neither situation is “respectively”. Accumulation was detected under both light regimes and at both times.

Response 54: Thank you for your useful comments. “respectively” has been deleted.

Point 55: ln 407. Change “Fffect” to “Effect”.

Response 55: Thank you for your useful comments. “Fffect” has been changed to “Effect”.

Point 56: ln 417-418. You do not need both “induced” and “upregulated”; “induced” is preferable here.

Response 56: Thank you for your useful comments. “upregulated” has been deleted.

Point 57: ln 422. Change “guidance” to “guide”.

Response 57: Thank you for your suggestion. “guidance” has been changed to “guide”.

Point 58: ln 430. Rephrase. The MLP28 does not resist virus infection, since a protein cannot be infected. MLP28 may help the host plant resist virus infection.

Response 58: Thank you for your suggestion. We have changed the sentence to “NbMLP28 could inhibit the infection of PVY to N. benthamiana.

Point 59: ln 443. Change “achieve” to “promote”. The virus achieves self-infection once it can replicate and move. Moreover, replication is achieved through other processes than the ones described here. The processes described in this work relate to promoting self-infection by inhibiting factors that are involved in the prevention of virus infection.

Response 59: Thank you for your suggestion. “achieve” has been changed to “promote”.

Point 60: ln 470. Change “and subsequently degrades” to “and is subsequently degraded”.

Response 60: Thank you for your suggestion. “and subsequently degrades” has been changed to “and is subsequently degraded”.

Point 61: ln 473. Insert “several” before “RNA viruses”. There may be member of some RNA virus families that are able to prevent this response.

Response 61: Thank you. As suggested, “several” has been inserted before “RNA viruses”.

Reviewer 2 Report

Dear Authors,

I have an opportunity to review manuscript entitled: “NbMLP43 ubiquitination and proteasomal degradation via the light responsive factor NbBBX24 to promote viral infection” submitted to Cells MDPI Journal.

Authors stated that they identified a novel major latex protein-like protein 43 (NbMLP43) that conferred resistance to Nicotiana benthamiana against potato virus Y (PVY) infection.

Authors presented the continuum of their results publicated in 2020 concentrated on  major latex protein-like protein 43 and resistance to PVY0. Authors revealed that PVY infection strongly induced NbMLP43 transcription but decreased NbMLP43 at the protein level; Moreover, Authors suggested that NbMLP43 interacted directly with NbBBX24—a light responsive factor as well as intermediate component targeting NbMLP43 for its ubiquitination and degradation thus ubiquitination-proteasome pathway.

The obtained results are quite interesting in the subject of plant resistance to PVY, but several aspect should be explained or improved in the manuscript:

·       It should be clearly underlined in the material and methods section, which direct isolate and virus strains were used by Author ( for example did Authors test all PVY strain ? I supposed not); therefore it should be clearly written that these obtained results are clear only for PVY 0 ( and here isolate number should be added); The same situation with CMV and used TMV;

·       Moreover, the reader did not have in duties analyzed Authors previous studies, therefore I warmly suggest explain deeply (in introduction), what was the previous effect on PVY ( we know also the statement ”enhanced resistance”; Besides of that, better explanation of general MLP subfamily we can find in discussion than in introduction part;

·       I suggest also to add more and recent information about resistance to PVY in the introduction – I found only one citation;

·       Please unified the fonts and font’s size in the whole manuscript – in current form it is an impression copy-paste of some paragraphs; Moreover, unifique to ‘N. benthamiana’ not ‘Benthamiana’;

·       NBT and DAB staining in material and methods need to have citations;

·       Figure 1 should be enlarged -in current form it is difficult to obtain information- in panel G after enlargement please add information that it is subcellular localisation only in epidermis;

·       In Figure 2 – Authors stated “. At 12 dpi, GFP signals were detected in the leaves of the entire  mlp43 system; however, in the wild-type, they were only noted in the veins, petioles, and a few leaves” – these information is completely lost in so small photos -please correct it; The same situation with statement “The number and size of the infected areas in mlp43 system leaves were significantly higher than those in the wild-type leaves (Fig. 2c)”;

·       Figure 2 D- based on what kind of quantifications Authors stated that: “NBT and DAB staining results indicated that mlp43 released more reactive oxygen than the wild-type, indirectly showing that the virus content in mlp43 was higher than that in wild-type (Fig. 2D)- Firstly – the photos had wrong markings, secondly, what was detected by DAB and what by NBT ?

In general, putting it mildly, the results are described in a laconic way;

Author Response

Dear Editor,

Thank you for your letter and the reviewers’ valuable comments concerning our manuscript: "NbMLP43 ubiquitination and proteasomal degradation via the light responsive factor NbBBX24 to promote viral infection" (Manuscript Number: cells-2169325). We have modified the manuscript accordingly, and respond to reviewers’ comments that are listed below. 

Academic Editor Comments: The authors need to explain more and better about the purpose of promoter analysis in Table 1 (It  was referred only once on page 5, and no further use even in discusion). They also need to show the positions of cis-acting regulatory elements in the 2.0 kb-length sequence (probably as a Supplementary Figure). 

Response: Thank you for your useful comments. Referring to published references, MLPs respond to biotic and abiotic stresses based on different signaling pathways, respectively. In this study, it was found that there are elements that respond to SA signals and stress by analyzing the promoter sequence of NbMLP43, and then follow-up studies on SA regulatory signals and NbMLP43 antiviral function was conducted.

Point 1: It should be clearly underlined in the material and methods section, which direct isolate and virus strains were used by Author ( for example did Authors test all PVY strain ? I supposed not); therefore it should be clearly written that these obtained results are clear only for PVY 0 ( and here isolate number should be added); The same situation with CMV and used TMV;

Response 1: We appreciate this observation. According to your suggestion, we have added the isolate number of PVY, CMV and TMV in Plant materials and viral strains parts. The PVY was PVYN:0 strain, the CMV was CMVⅠB strain, and the TMV was TMV U1 strain.

Point 2: Moreover, the reader did not have in duties analyzed Authors previous studies, therefore I warmly suggest explain deeply (in introduction), what was the previous effect on PVY ( we know also the statement ”enhanced resistance”; Besides of that, better explanation of general MLP subfamily we can find in discussion than in introduction part;

  • I suggest also to add more and recent information about resistance to PVY in the introduction – I found only one citation;

Response 2: Thank you for your useful comments. The change has been made accordingly. But the reference on MLP resistant to PVY is currently only this one.

Point 3: Please unified the fonts and font’s size in the whole manuscript – in current form it is an impression copy-paste of some paragraphs; Moreover, unifique to ‘N. benthamiana’ not ‘Benthamiana’;

Response 3: Thank you for your suggestion. We have unified the fonts and font’s size in the whole manuscript. Moreover, “N. benthamiana” was unifique in the manuscript.

Point 4: NBT and DAB staining in material and methods need to have citations;

Response 4: Thank you for your suggestion. We have inserted citations [27,28].

Point 5: Figure 1 should be enlarged -in current form it is difficult to obtain information- in panel G after enlargement please add information that it is subcellular localisation only in epidermis;

Response 5: Thank you. The change has been made accordingly.

Point 6: In Figure 2 – Authors stated “. At 12 dpi, GFP signals were detected in the leaves of the entire  mlp43 system; however, in the wild-type, they were only noted in the veins, petioles, and a few leaves” – these information is completely lost in so small photos -please correct it; The same situation with statement “The number and size of the infected areas in mlp43 system leaves were significantly higher than those in the wild-type leaves (Fig. 2c)”;

Response 6: Thank you for your useful comments. The change has been made accordingly.

Point 7: Figure 2 D- based on what kind of quantifications Authors stated that: “NBT and DAB staining results indicated that mlp43 released more reactive oxygen than the wild-type, indirectly showing that the virus content in mlp43 was higher than that in wild-type (Fig. 2D)- Firstly – the photos had wrong markings, secondly, what was detected by DAB and what by NBT ? In general, putting it mildly, the results are described in a laconic way

Response 7: Thank you for your useful comments. The change has been made accordingly and Fig. 2D has been revised.

Round 2

Reviewer 2 Report

Dear Editors,

Unfortunately, Authors did not take into account all my suggestion:

- they claim to have done as suggested, but that is not the case at all;

- there are still lack of quantification methods for  DAB and NBT analyses and we still can find statements like : "NBT and DAB staining results indicated that mlp43 released more ROS than the wild-type, indirectly showing that the virus content in mlp43 was higher than that in wild-type (Fig. 2D) - FOR SURE figure 2D did not inform the reader about these statements;

Moreover, methods with diaminobenzidine did not inform as about peroxidase activity - like it is still stated in line 317;

-furthermore, I ask Authors about give the reader more information about resistance reaction to PVY - but I did not find any new information in introduction;

Therefore, I don't change my decision;

Sincerely

Author Response

Dear Editor,    

Thank you for your letter and the reviewers’ valuable comments concerning our manuscript: "NbMLP43 ubiquitination and proteasomal degradation via the light responsive factor NbBBX24 to promote viral infection" (Manuscript Number: cells-2169325). We have modified the manuscript accordingly, and respond to reviewers’ comments that are listed below.

Academic Editor Comments: The authors need to explain more and better about the purpose of promoter analysis in Table 1 (It  was referred only once on page 5, and no further use even in discusion). They also need to show the positions of cis-acting regulatory elements in the 2.0 kb-length sequence (probably as a Supplementary Figure). 

Response: Thank you for your useful comments. Referring to published references, MLPs respond to biotic and abiotic stresses based on different signaling pathways, respectively. In this study, it was found that there are elements that respond to SA signals and stress by analyzing the promoter sequence of NbMLP43, and then follow-up studies on SA regulatory signals and NbMLP43 antiviral function was conducted.

Point 1: there are still lack of quantification methods for DAB and NBT analyses and we still can find statements like : "NBT and DAB staining results indicated that mlp43 released more ROS than the wild-type, indirectly showing that the virus content in mlp43 was higher than that in wild-type (Fig. 2D) - FOR SURE figure 2D did not inform the reader about these statements;

- Moreover, methods with diaminobenzidine did not inform as about peroxidase activity - like it is still stated in line 317;

Response 1: Thank you for your useful comments. We changed the description of Fig. 2D in Results 3.3 and added detection targets of DAB and NBT in the Methods section.

Point 2: -furthermore, I ask Authors about give the reader more information about resistance reaction to PVY - but I did not find any new information in introduction;

Response 2: Sorry for not understanding your meaning correctly, we have added three references on PVY resistance now.

Round 3

Reviewer 2 Report

In my opinion after the second revision manuscript was significantly improved and almost all my suggestions are taking into account. Therefore, I suggest accept manuscript in current form,

Sincerely